# Think Globally, Act Locally: A Deep Neural Network Approach to High-Dimensional Time Series Forecasting

Rajat Sen[1], Hsiang-Fu Yu[1], and Inderjit Dhillon[2]

[1]Amazon
[2]Amazon and UT Austin

## Abstract

Forecasting high-dimensional time series plays a crucial role in many applications such as demand forecasting and financial predictions. Modern datasets can have millions of correlated time-series that evolve together, i.e they are extremely high dimensional (one dimension for each individual time-series). There is a need for exploiting global patterns and coupling them with local calibration for better prediction. However, most recent deep learning approaches in the literature are one-dimensional, i.e, even though they are trained on the whole dataset, during prediction, the future forecast for a single dimension mainly depends on past values from the same dimension. In this paper, we seek to correct this deficiency and propose DeepGLO, a deep forecasting model which *thinks globally and acts locally*. In particular, DeepGLO is a hybrid model that combines a *global* matrix factorization model regularized by a temporal convolution network, along with another temporal network that can capture *local* properties of each time-series and associated covariates. Our model can be trained effectively on high-dimensional but diverse time series, where different time series can have vastly different scales, *without* a priori normalization or rescaling. Empirical results demonstrate that DeepGLO can outperform state-of-the-art approaches; for example, we see more than 25% improvement in WAPE over other methods on a public dataset that contains more than 100K-dimensional time series.

## 1 Introduction

Time-series forecasting is an important problem with many industrial applications like retail demand forecasting [21], financial predictions [15], predicting traffic or weather patterns [5]. In general it plays a key role in automating business processes [17]. Modern data-sets can have millions of correlated time-series over several thousand time-points. For instance, in an online shopping portal like Amazon or Walmart, one may be interested in the future daily demands for all items in a category, where the number of items may be in millions. This leads to a problem of forecasting $n$ time-series (one for each of the $n$ items), given past demands over $t$ time-steps. Such a time series data-set can be represented as a matrix $\mathbf{Y} \in \mathbb{R}^{n \times t}$ and we are interested in the *high-dimensional* setting where $n$ can be of the order of millions.

Traditional time-series forecasting methods operate on individual time-series or a small number of time-series at a time. These methods include the well known AR, ARIMA, exponential smoothing [19], the classical Box-Jenkins methodology [4] and more generally the linear state-space models [13]. However, these methods are not easily scalable to large data-sets with millions of time-series, owing to the need for individual training. Moreover, they cannot benefit from shared temporal patterns in the whole data-set while training and prediction.

Deep networks have gained popularity in time-series forecasting recently, due to their ability to model non-linear temporal patterns. Recurrent Neural Networks (RNN's) [10] have been popular in

sequential modeling, however they suffer from the gradient vanishing/exploding problems in training. Long Short Term Memory (LSTM) [11] networks alleviate that issue and have had great success in langulage modeling and other seq-to-seq tasks [11, 22]. Recently, deep time-series models have used LSTM blocks as internal components [9, 20]. Another popular architecture, that is competitive with LSTM's and arguably easier to train is temporal convolutions/causal convolutions popularized by the wavenet model [24]. Temporal convolutions have been recently used in time-series forecasting [3, 2]. These deep network based models can be trained on large time-series data-sets as a whole, in mini-batches. However, they still have two important shortcomings.

Firstly, most of the above deep models are difficult to train on data-sets that have wide *variation in scales* of the individual time-series. For instance in the item demand forecasting use-case, the demand for some popular items may be orders of magnitude more than those of niche items. In such data-sets, each time-series needs to be appropriately normalized in order for training to succeed, and then the predictions need to be scaled back to the original scale. The mode and parameters of normalization are difficult to choose and can lead to different accuracies. For example, in [9, 20] each time-series is whitened using the corresponding empirical mean and standard deviation, while in [3] the time-series are scaled by the corresponding value on the first time-point.

Secondly, even though these deep models are trained on the entire data-set, during prediction the models only focus on *local* past data i.e only the past data of a time-series is used for predicting the future of that time-series. However, in most datasets, *global* properties may be useful during prediction time. For instance, in stock market predictions, it might be beneficial to look at the past values of Alphabet, Amazon's stock prices as well, while predicting the stock price of Apple. Similarly, in retail demand forecasting, past values of similar items can be leveraged while predicting the future for a certain item. To this end, in [16], the authors propose a combination of 2D convolution and recurrent connections, that can take in multiple time-series in the input layer thus capturing global properties during prediction. However, this method does not scale beyond a few thousand time-series, owing to the growing size of the input layer. On the other end of the spectrum, TRMF [29] is a temporally regularized matrix factorization model that can express all the time-series as linear combinations of *basis time-series*. These basis time-series can capture *global* patterns during prediction. However, TRMF can only model linear temporal dependencies. Moreover, there can be approximation errors due to the factorization, which can be interpreted as a lack of local focus i.e the model only concentrates on the global patterns during prediction.

In light of the above discussion, we aim to propose a deep learning model that can *think globally and act locally* i.e., leverage both *local and global* patterns during training and prediction, and also can be trained reliably even when there are *wide variations in scale*. The **main contributions** of this paper are as follows:

- In Section A, we discuss issues with wide variations in scale among different time-series, and propose a simple initialization scheme, LeveledInit for Temporal Convolution Networks (TCN) that enables training without apriori normalization.
- In Section 5.1, we present a matrix factorization model regularized by a TCN (TCN-MF), that can express each time-series as linear combination of $k$ basis time-series, where $k$ is much less than the number of time-series. Unlike TRMF, this model can capture non-linear dependencies as the regularization and prediction is done using a temporal convolution trained concurrently and also is amicable to scalable mini-batch training. This model can handle *global* dependencies during prediction.
- In Section 5.2, we propose DeepGLO, a *hybrid* model, where the predictions from our *global* TCN-MF model, is provided as covariates for a temporal convolution network, thereby enabling the final model to focus both on local per time-series properties as well as global dataset wide properties, while both training and prediction.
- In Section 6, we show that DeepGLO outperforms other benchmarks on four real world time-series data-sets, including a public wiki dataset which contains more than $110K$ dimensions of time series. More details can be found in Tables 1 and 2.

## 2 Related Work

The literature on time-series forecasting is vast and spans several decades. Here, we will mostly focus on recent deep learning approaches. For a comprehensive treatment of traditional methods, we refer the readers to [13, 19, 4, 18, 12] and the references there in.

In recent years deep learning models have gained popularity in time-series forecasting. DeepAR [9] proposes a LSTM based model where parameters of Bayesian models for the future time-steps are predicted as a function of the corresponding hidden states of the LSTM. In [20], the authors combine linear state space models with deep networks. In [26], the authors propose a time-series model where all history of a time-series is encoded using an LSTM block, and a multi horizon MLP decoder is used to decode the input into future forecasts. LSTNet [16] can leverage correlations between multiple time-series through a combination of 2D convolution and recurrent structures. However, it is difficult to scale this model beyond a few thousand time-series because of the growing size of the input layer. Temporal convolutions have been recently used for time-series forecasting [3].

Matrix factorization with temporal regularization was first used in [27] in the context of speech de-noising. A spatio-temporal deep model for traffic data has been proposed in [28]. Perhaps closest to our work is TRMF [29], where the authors propose an AR based temporal regularization. In this paper, we extend this work to non-linear settings where the temporal regularization can be performed by a temporal convolution network (see Section 4). We further combine the global matrix factorization model with a temporal convolution network, thus creating a hybrid model that can focus on both local and global properties. There has been a concurrent work [25], where an RNN has been used to evolve a global state common to all time-series.

## 3    Problem Setting

We consider the problem of forecasting high-dimensional time series over future time-steps. High-dimensional time-series datasets consist of several possibly correlated time-series evolving over time along with corresponding covariates, and the task is to forecast the values of those time-series in future time-steps. Before, we formally define the problem, we will set up some notation.

**Notation:**  We will use bold capital letters to denote matrices such as $\mathbf{M} \in \mathbb{R}^{m \times n}$. $M_{ij}$ and $\mathbf{M}[i, j]$ will be used interchangeably to denote the $(i, j)$-th entry of the matrix $\mathbf{M}$. Let $[n] \triangleq \{1, 2, ..., n\}$ for a positive integer $n$. For $\mathcal{I} \subseteq [m]$ and $\mathcal{J} \subseteq [n]$, the notation $\mathbf{M}[\mathcal{I}, \mathcal{J}]$ will denote the sub-matrix of $\mathbf{M}$ with rows in $\mathcal{I}$ and columns in $\mathcal{J}$. $\mathbf{M}[:, \mathcal{J}]$ means that all the rows are selected and similarly $\mathbf{M}[\mathcal{I}, :]$ means all the columns are chosen. $\mathcal{J} + s$ denotes all the set of elements in $\mathcal{J}$ increased by $s$. The notation $i : j$ for positive integers $j > i$, is used to denote the set $\{i, i + 1, ..., j\}$. $\|\mathbf{M}\|_F$, $\|\mathbf{M}\|_2$ denote the Frobenius and Spectral norms respectively. We will also define 3-dimensional tensor notation in a similar way as above. Tensors will also be represented by bold capital letters $\mathbf{T} \in \mathbb{R}^{m \times r \times n}$. $T_{ijk}$ and $\mathbf{T}[i, j, k]$ will be used interchangeably to denote the $(i, j, k)$-th entry of the tensor $\mathbf{T}$. For $\mathcal{I} \subseteq [m]$, $\mathcal{J} \subseteq [n]$ and $\mathcal{K} \subseteq [r]$, the notation $\mathbf{T}[\mathcal{I}, \mathcal{K}, \mathcal{J}]$ will denote the sub-tensor of $\mathbf{T}$, restricted to the selected coordinates. By convention, all vectors in this paper are row vectors unless otherwise specified. $\|\mathbf{v}\|_p$ denotes the $p$-norm of the vector $\mathbf{v} \in \mathbb{R}^{1 \times n}$. $\mathbf{v}_{\mathcal{I}}$ denotes the sub-vector with entries $\{v_i : \forall i \in \mathcal{I}\}$ where $v_i$ denotes the $i$-th coordinate of $\mathbf{v}$ and $\mathcal{I} \subseteq [n]$. The notation $\mathbf{v}_{i:j}$ will be used to denote the vector $[v_i, ..., v_j]$. The notation $[\mathbf{v}; \mathbf{u}] \in \mathbb{R}^{1 \times 2n}$ will be used to denote the concatenation of two row vectors $\mathbf{v}$ and $\mathbf{u}$. For a vector $\mathbf{v} \in \mathbb{R}^{1 \times n}$, $\mu(\mathbf{v}) \triangleq (\sum_i v_i)/n$ denotes the empirical mean of the coordinates and $\sigma(\mathbf{v}) \triangleq \sqrt{(\sum_i (v_i - \mu(\mathbf{v})^2))/n}$ denotes the empirical standard deviation.

**Forecasting Task:**  A time-series data-set consists of the raw time-series, represented by a matrix $\mathbf{Y} = [\mathbf{Y}^{(\text{tr})}\mathbf{Y}^{(\text{te})}]$, where $\mathbf{Y}^{(\text{tr})} \in \mathbb{R}^{n \times t}$, $\mathbf{Y}^{(\text{te})} \in \mathbb{R}^{n \times \tau}$, $n$ is the number of time-series, $t$ is the number time-points observed during training phase, $\tau$ is the window size for forecasting. $\mathbf{y}^{(i)}$ is used to denote the $i$-th time series, i.e., the $i$-th row of $\mathbf{Y}$. In addition to the raw time-series, there may optionally be observed covariates, represented by the tensor $\mathbf{Z} \in \mathbb{R}^{n \times r \times (t+\tau)}$. $\mathbf{z}_j^{(i)} = \mathbf{Z}[i, :, j]$ denotes the $r$-dimensional covariates for time-series $i$ and time-point $j$. Here, the covariates can consist of global features like time of the day, day of the week etc which are common to all time-series, as well as covariates particular to each time-series, for example vectorized text features describing each time-series. The forecasting task is to accurately predict the future in the test range, given the original time-series $\mathbf{Y}^{(\text{tr})}$ in the training time-range and the covariates $\mathbf{Z}$. $\hat{\mathbf{Y}}^{(\text{te})} \in \mathbb{R}^{n \times \tau}$ will be used to denote the predicted values in the test range.

**Objective:**  The quality of the predictions are generally measured using a metric calculated between the predicted and actual values in the test range. One of the popular metrics is the normalized absolute

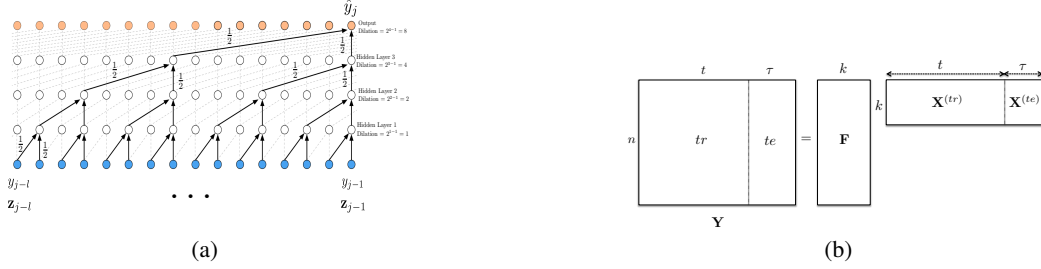

|          | (a)          |          | (b)          |

Figure 1: Fig. 1a contains an illustration of a TCN. Note that the base image has been borrowed from [24]. The network has $d = 4$ layers, with filter size $k = 2$. The network maps the input $\mathbf{y}_{t-l:t-1}$ to the one-shifted output $\hat{\mathbf{y}}_{t-l+1:t}$. Figure 1b presents an illustration of the matrix factorization approach in time-series forecasting. The $\mathbf{Y}^{(\text{tr})}$ training matrix can be factorized into low-rank factors $\mathbf{F}$ ($\in \mathbb{R}^{n \times k}$) and $\mathbf{X}^{(\text{tr})}$ ($\in \mathbb{R}^{k \times t}$). If $\mathbf{X}^{(\text{tr})}$ preserves temporal structures then the future values $\mathbf{X}^{(\text{te})}$ can be predicted by a time-series forecasting model and thus the test period predictions can be made as $\mathbf{F}\mathbf{X}^{(\text{te})}$.

deviation metric [29], defined as follows,

$$\mathcal{L}(Y^{(\text{obs})}, Y^{(\text{pred})}) = \frac{\sum_{i=1}^{n} \sum_{j=1}^{\tau} |Y_{ij}^{(\text{obs})} - Y_{ij}^{(\text{pred})}|}{\sum_{i=1}^{n} \sum_{j=1}^{\tau} |Y_{ij}^{(\text{obs})}|}, \tag{1}$$

where $Y^{(\text{obs})}$ and $Y^{(\text{pred})}$ are the observed and predicted values, respectively. This metric is also referred to as WAPE in the forecasting literature. Other commonly used evaluation metrics are defined in Section C.2. Note that (1) is also used as a loss function in one of our proposed models. We also use the squared-loss $\mathcal{L}_2(Y^{(\text{obs})}, Y^{(\text{pred})}) = (1/n\tau) \left\| Y^{(\text{obs})} - Y^{(\text{pred})} \right\|_F^2$, during training.

# 4 LeveledInit: Handling Diverse Scales with TCN

In this section, we propose LeveledInit a simple initialization scheme for Temporal Convolution Networks (TCN) [2] which is designed to handle high-dimensional time-series data with wide variation in scale, without apriori normalization. As mentioned before, deep networks are difficult to train on time-series datasets, where the individual time-series have diverse scales. LSTM based models cannot be reliably trained on such datasets and may need apriori standard normalization [16] or pre-scaling of the bayesian mean predictions [9]. TCN's have also been shown to require apriori normalization [3] for time-series predictions. The choice of normalization parameters can have a significant effect on the prediction performance. Here, we show that a simple initialization scheme for the TCN network weights can alleviate this problem and lead to reliable training without apriori normalization. First, we will briefly discuss the TCN architecture.

**Temporal Convolution:** Temporal convolution (also referred to as Causal Convolution) [24, 3, 2] are multi-layered neural networks comprised of 1D convolutional layers with dilation. The dilation in layer $i$ is usually set as $\text{dil}(i) = 2^{i-1}$. A temporal convolution network with filter size $k$ and number of layers $d$ has a dynamic range (or look-back) of $l' := 1 + l = 1 + 2(k-1)2^{d-1}$. Note that it is assumed that the stride is 1 in every layer and layer $i$ needs to be zero-padded in the beginning with $(k-1)\text{dil}(i)$ zeros. An example of a temporal convolution network with one channel per layer is shown in Fig. 1a. For more details, we refer the readers to the general description in [2]. Note that in practice, we can have multiple channels per layer of a TC network. The TC network can thus be treated as an object that takes in the previous values of a time-series $\mathbf{y}_{\mathcal{J}}$, where $\mathcal{J} = \{j - l, j - l + 1, \cdots, j - 1\}$ along with the past covariates $\mathbf{z}_{\mathcal{J}}$, corresponding to that time-series and outputs the one-step look ahead predicted value $\hat{\mathbf{y}}_{\mathcal{J}+1}$. We will denote a temporal convolution neural network by $\mathcal{T}(\cdot|\Theta)$, where $\Theta$ is the parameter weights in the temporal convolution network. Thus, we have $\hat{\mathbf{y}}_{\mathcal{J}+1} = \mathcal{T}(\mathbf{y}_{\mathcal{J}}, \mathbf{z}_{\mathcal{J}}|\Theta)$. The same operators can be defined on matrices. Given $\mathbf{Y} \in \mathbb{R}^{n \times t}, \mathbf{Z} \in \mathbb{R}^{n \times r \times (t+\tau)}$ and a set of row indices $\mathcal{I} = \{i_1, ..., i_{b_n}\} \subset [n]$, we can write $\hat{\mathbf{Y}}[\mathcal{I}, \mathcal{J}+1] = \mathcal{T}(\mathbf{Y}[\mathcal{I}, \mathcal{J}], \mathbf{Z}[\mathcal{I}, :, \mathcal{J}]|\Theta)$.

LeveledInit **Scheme:** One possible method to alleviate the issues with diverse scales, is to start with initial network parameters, that results in approximately predicting the average value of a given window of past time-points $\mathbf{y}_{j-l:j-1}$, as the future prediction $\hat{y}_j$. The hope is that, over the course of training, the network would learn to predict variations around this mean prediction, given that the variations around this mean is relatively scale free. This can be achieved through a simple initialization

scheme for some configurations of TCN, which we call LeveledInit. For ease of exposition, let us consider the setting without covariates and only one channel per layer, which can be functionally represented as $\hat{\mathbf{y}}_{\mathcal{J}+1} = \mathcal{T}(\mathbf{y}_{\mathcal{J}}|\Theta)$. In this case, the initialization scheme is to simply set all the filter weights to $1/k$, where $k$ is the filter size in every layer. This results in a proposition.

**Proposition 1** (LeveledInit). *Let $\mathcal{T}(\cdot|\Theta)$ be a TCN with one channel per layer, filter size $k = 2$, number of layers d. Here $\Theta$ denotes the weights and biases of the network. Let $[\hat{y}_{j-l+1}, \cdots, \hat{y}_j] :=$ $\mathcal{T}(\mathbf{y}_{\mathcal{J}}|\Theta)$ for $\mathcal{J} = \{j - l, ...., j - 1\}$ and $l = 2(k - 1)2^{d-1}$. If all the biases in $\Theta$ are set to 0 and all the weights set to $1/k$ then, $\hat{y}_j = \mu(\mathbf{y}_{\mathcal{J}})$ if $\mathbf{y} \geq \mathbf{0}$ and all activation functions are ReLUs.*

The above proposition shows that LeveledInit results in a prediction $\hat{y}_j$, which is the average of the past $l$ time-points, where $l$ is the dynamic range of the TCN, when filter size is $k = 2$ (see Fig. 1a). The proof of proposition 1 (see Section A)) follows from the fact that an activation value in an internal layer is the average of the corresponding $k$ inputs from the previous layer, and an induction on the layers yields the results. LeveledInit can also be extended to the case with covariates, by setting the corresponding filter weights to 0 in the input layer. It can also be easily extended to multiple channels per layer with $k = 2$. In Section 6, we show that a TCN with LeveledInit can be trained reliably without apriori normalization, on real world datasets, even for values of $k \neq 2$. We provide the psuedo-code for training a TCN with LeveledInit as Algorithm 1.

Note that we have also experimented with a more sophisticated variation of the Temporal Convolution architecture termed as Deep Leveled Network (DLN ), which we include in Appendix A. However, we observed that the simple initialization scheme for TCN (LeveledInit) can match the performance of the Deep Leveled network.

## 5    DeepGLO: A Deep Global Local Forecaster

In this section we will introduce our hybrid model DeepGLO, that can leverage both global and local features, during training and prediction. In Section 5.1, we present the global component, TCN regularized Matrix Factorization (TCN-MF). This model can capture global patterns during prediction, by representing each of the original time-series as a linear combination of $k$ basis time-series, where $k \ll n$. In Section 5.2, we detail how the output of the global model can be incorporated as an input covariate dimension for a TCN, thus leading to a hybrid model that can both focus on local per time-series signals and leverage global dataset wide components.

---

**Algorithm 1** Mini-batch Training for TCN with LeveledInit

**Require:** learning rate $\eta$, horizontal batch size $b_t$, vertical batch size $b_n$, and `maxiters`
1: Initialize $\mathcal{T}(\cdot|\Theta)$ according to LeveledInit
2: **for** iter $= 1, \cdots,$ `maxiters` **do**
3:     **for** each batch with indices $\mathcal{I}$ and $\mathcal{J}$ in an epoch **do**
4:         $\mathcal{I} = \{i_1, ..., i_{b_n}\}$ and $\mathcal{J} = \{j + 1, j + 2, ..., j + b_t\}$
5:         $\hat{\mathbf{Y}} \leftarrow \mathcal{T}(\mathbf{Y}[\mathcal{I}, \mathcal{J}], \mathbf{Z}[\mathcal{I}, :, \mathcal{J}]|\Theta)$
6:         $\Theta \leftarrow \Theta - \eta \frac{\partial}{\partial \Theta} \mathcal{L}(\mathbf{Y}[\mathcal{I}, \mathcal{J} + 1], \hat{\mathbf{Y}})$
7:     **end for**
8: **end for**

---

**Algorithm 2** Temporal Matrix Factorization Regularized by TCN (TCN-MF)

**Require:** $\text{iters}_{init}, \text{iters}_{train}, \text{iters}_{alt}$.
1: /* *Model Initialization* */
2: Initialize $\mathcal{T}_X(\cdot)$ by LeveledInit
3: Initialize $\mathbf{F}$ and $\mathbf{X}^{(tr)}$ by Alg 3 for $\text{iters}_{init}$ iterations.
4: /* *Alternate training cycles* */
5: **for** iter $= 1, 2, ..., \text{iters}_{alt}$ **do**
6:     Update $\mathbf{F}$ and $\mathbf{X}^{(tr)}$ by Alg 3 for $\text{iters}_{train}$ iterations
7:     Update $\mathcal{T}_X(\cdot)$ by Alg 1 on $\mathbf{X}^{(tr)}$ for $\text{iters}_{train}$ iterations, with *no covariates*.
8: **end for**

---

### 5.1    Global: Temporal Convolution Network regularized Matrix Factorization (TCN-MF)

In this section we propose a low-rank matrix factorization model for time-series forecasting that uses a TCN for regularization. The idea is to factorize the training time-series matrix $\mathbf{Y}^{(tr)}$ into low-rank factors $\mathbf{F} \in \mathbb{R}^{n \times k}$ and $\mathbf{X}^{(tr)} \in \mathbb{R}^{k \times t}$, where $k \ll n$. This is illustrated in Figure 1b. Further, we would like to encourage a temporal structure in $\mathbf{X}^{(tr)}$ matrix, such that the future values $\mathbf{X}^{(te)}$ in the test range can also be forecasted. Let $\mathbf{X} = [\mathbf{X}^{(tr)}\mathbf{X}^{(te)}]$. Thus, the matrix $\mathbf{X}$ can be thought of to be comprised of $k$ *basis time-series* that capture the global temporal patterns in the whole data-set and all the original time-series are linear combinations of these basis time-series. In the next subsection we will describe how a TCN can be used to encourage the temporal structure for $\mathbf{X}$.

**Temporal Regularization by a TCN:** If we are provided with a TCN that captures the temporal patterns in the training data-set $\mathbf{Y}^{(tr)}$, then we can encourage temporal structures in $\mathbf{X}^{(tr)}$ using this model. Let us assume that the said network is $\mathcal{T}_X(\cdot)$. The temporal patterns can be encouraged by

including the following temporal regularization into the objective function:

$$\mathcal{R}(\mathbf{X}^{(\text{tr})} \mid \mathcal{T}_X(\cdot)) := \frac{1}{|\mathcal{J}|} \mathcal{L}_2\left(\mathbf{X}[:, \mathcal{J}], \mathcal{T}_X\left(\mathbf{X}[:, \mathcal{J} - 1]\right)\right), \tag{2}$$

where $\mathcal{J} = \{2, \cdots, t\}$ and $\mathcal{L}_2(\cdot, \cdot)$ is the squared-loss metric, defined before. This implies that the values of the $\mathbf{X}^{(tr)}$ on time-index $j$ are close to the predictions from the temporal network applied on the past values between time-steps $\{j - l, ..., j - 1\}$. Here, $l + 1$ is equal to the dynamic range of the network defined in Section 4. Thus the overall loss function for the factors and the temporal network is as follows:

$$\mathcal{L}_G(\mathbf{Y}^{(\text{tr})}, \mathbf{F}, \mathbf{X}^{(\text{tr})}, \mathcal{T}_X) := \mathcal{L}_2\left(\mathbf{Y}^{(\text{tr})}, \mathbf{F}\mathbf{X}^{(\text{tr})}\right) + \lambda_{\mathcal{T}} \mathcal{R}(\mathbf{X}^{(\text{tr})} \mid \mathcal{T}_X(\cdot)), \tag{3}$$

where $\lambda_{\mathcal{T}}$ is the regularization parameter for the temporal regularization component.

**Training:** The low-rank factors $\mathbf{F}, \mathbf{X}^{(\text{tr})}$ and the temporal network $\mathcal{T}_X(\cdot)$ can be trained alternatively to approximately minimize the loss in Eq. (3). The overall training can be performed through mini-batch SGD and can be broken down into two main components performed alternatingly: $(i)$ given a fixed $\mathcal{T}_X(\cdot)$ minimize $\mathcal{L}_G(\mathbf{F}, \mathbf{X}^{(\text{tr})}, \mathcal{T}_X)$ with respect to the factors $\mathbf{F}, \mathbf{X}^{(\text{tr})}$ - Algorithm 3 and $(ii)$ train the network $\mathcal{T}_X(\cdot)$ on the matrix $\mathbf{X}^{(\text{tr})}$ using Algorithm 1.

The overall algorithm is detailed in Algorithm 2. The TCN $\mathcal{T}_X(\cdot)$ is first initialized by LeveledInit. Then in the second initialization step, two factors $\mathbf{F}$ and $\mathbf{X}^{(\text{tr})}$ are trained using the initialized $\mathcal{T}_X(\cdot)$ (step 3), for iters$_{\text{init}}$ iterations. This is followed by the iters$_{\text{alt}}$ alternative steps to update $\mathbf{F}$, $\mathbf{X}^{(\text{tr})}$, and $\mathcal{T}_X(\cdot)$ (steps 5-7).

**Prediction:** The trained model local network $\mathcal{T}_X(\cdot)$ can be used for multi-step look-ahead prediction in a standard manner. Given the past data-points of a basis time-series, $\mathbf{x}_{j-l:j-1}$, the prediction for the next time-step, $\hat{x}_j$ is given by $[\hat{x}_{j-l+1}, \cdots, \hat{x}_j] := \mathcal{T}_X(\mathbf{x}_{j-l:j-1})$ Now, the one-step look-ahead prediction can be concatenated with the past values to form the sequence $\tilde{\mathbf{x}}_{j-l+1:j} = [\mathbf{x}_{j-l+1:j-1}\hat{x}_j]$, which can be again passed through the network to get the next prediction: $[\cdots, \hat{x}_{j+1}] = \mathcal{T}_X(\tilde{\mathbf{x}}_{j-l+1:j})$. The same procedure can be repeated $\tau$ times to predict $\tau$ time-steps ahead in the future. Thus we can obtain the basis time-series in the test-range $\hat{\mathbf{X}}^{(te)}$. The final global predictions are then given by $\mathbf{Y}^{(te)} = \mathbf{F}\hat{\mathbf{X}}^{(te)}$.

**Remark 1.** *Note that* TCN-MF *can perform rolling predictions without retraining unlike* TRMF. *We provide more details in Appendix B, in the interest of space.*

---

**Algorithm 3** Training the Low-rank factors $\mathbf{F}, \mathbf{X}^{(\text{tr})}$ given a fixed network $\mathcal{T}_X(\cdot)$, for one epoch

---

**Require:** learning rate $\eta$, a TCN $\mathcal{T}_X(\cdot)$.
1: **for** each batch with indices $\mathcal{I}$ and $\mathcal{J}$ in an epoch **do**
2: $\quad \mathcal{I} = \{i_1, ..., i_{b_n}\}$ and $\mathcal{J} = \{j + 1, j + 2, ..., j + b_t\}$
3: $\quad \mathbf{X}[:, \mathcal{J}] \leftarrow \mathbf{X}[:, \mathcal{J}] - \eta \frac{\partial}{\partial \mathbf{X}[:, \mathcal{J}]} \mathcal{L}_G(\mathbf{Y}[\mathcal{I}, \mathcal{J}], \mathbf{F}[\mathcal{I}, :], \mathbf{X}[:, \mathcal{J}], \mathcal{T}_X)$
4: $\quad \mathbf{F}[\mathcal{I}, :] \leftarrow \mathbf{F}[\mathcal{I}, :] - \eta \frac{\partial}{\partial \mathbf{F}[\mathcal{I}, :]} \mathcal{L}_G(\mathbf{Y}[\mathcal{I}, \mathcal{J}], \mathbf{F}[\mathcal{I}, :], \mathbf{X}[:, \mathcal{J}], \mathcal{T}_X)$
5: **end for**

---

**Algorithm 4** DeepGLO- Deep Global Local Forecaster

---

1: Obtain global $\mathbf{F}$, $\mathbf{X}^{(\text{tr})}$ and $\mathcal{T}_X(\cdot)$ by Alg 2.
2: Initialize $\mathcal{T}_Y(\cdot)$ with number of inputs $r + 2$ and LeveledInit.
3: /* *Training hybrid model* */
4: Let $\hat{\mathbf{Y}}^{(g)}$ be the global model prediction in the training range.
5: Create covariates $\mathbf{Z}' \in \mathbb{R}^{n \times (r+1) \times t}$ s.t $\mathbf{Z}'[:, 1, :] = \hat{\mathbf{Y}}^{(g)}$ and $\mathbf{Z}'[:, 2 : r + 1, :] = \mathbf{Z}[:, :, 1 : t]$.
6: Train $\mathcal{T}_Y(\cdot)$ using Algorithm 1 with time-series $\mathbf{Y}^{(tr)}$ and covariates $\mathbf{Z}'$.

---

### 5.2 Combining the Global Model with Local Features

In this section, we present our final hybrid model. Recall that our forecasting task has as input the training raw time-series $\mathbf{Y}^{(tr)}$ and the covariates $\mathbf{Z} \in \mathbb{R}^{n \times r \times (t+\tau)}$. Our hybrid forecaster is a TCN $\mathcal{T}_Y(\cdot|\Theta_Y)$ with a input size of $r + 2$ dimensions: $(i)$ one of the inputs is reserved for the past points of the original raw time-series, $(ii)$ $r$ inputs for the original $r$-dimensional covariates and $(iii)$ the remaining dimension is reserved for the output of the global TCN-MF model, which is fed as input covariates. The overall model is demonstrated in Figure 2. The training pseudo-code for our model is provided as Algorithm 4.

Therefore, by providing the global TCN-MF model prediction as covariates to a TCN, we can make the final predictions a function of global dataset wide properties as well as the past values of the local time-series and covariates. Note that both rolling predictions and multi-step look-ahead predictions can be performed by DeepGLO, as the global TCN-MF model and the hybrid TCN $\mathcal{T}_Y(\cdot)$ can

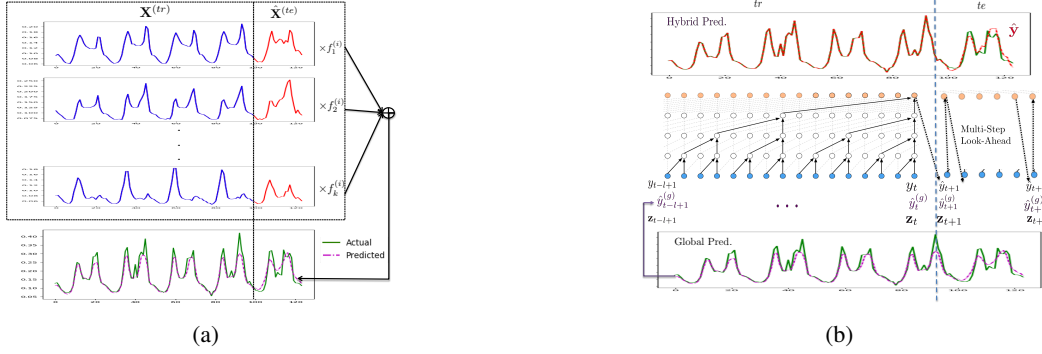

|     | (a)     | (b)     |
|-----|---------|---------|

Figure 2: In Fig. 2a, we show some of the basis time-series extracted from the traffic dataset, which can be combined linearly to yield individual original time-series. It can be seen that the basis series are highly temporal and can be predicted in the test range using the network $\mathcal{T}_X(\cdot|\Theta_X)$. In Fig. 2b (base image borrowed from [24]), we show an illustration of DeepGLO. The TCN shown is the network $\mathcal{T}_Y(\cdot)$, which takes in as input the original time-points, the original covariates and the output of the global model as covariates. Thus this network can combine the local properties with the output of the global model during prediction.

Table 1: Data statistics and Evaluation settings. In the rolling Pred., $\tau_w$ denotes the number of time-points in each window and $n_w$ denotes the number of rolling windows. $\text{std}(\{\mu\})$ denotes the standard deviation among the means of all the time series in the data-set i.e $\text{std}(\{\mu\}) = \text{std}(\{\mu(\mathbf{y}^{(i)})\}_{i=1}^n)$. Similarly, $\text{std}(\{\text{std}\})$ denotes the standard deviation among the std. of all the time series in the data-set i.e $\text{std}(\{\text{std}\}) = \text{std}(\{\text{std}(\mathbf{y}^{(i)})\}_{i=1}^n)$. It can be seen that the electricity and wiki datasets have wide variations in scale.

| Data | $n$ | $t$ | $\tau_w$ | $n_w$ | $\text{std}(\{\mu(\mathbf{y}_i)\})$ | $\text{std}(\{\text{std}(\mathbf{y}_i)\})$ |
|------|-----|-----|----------|-------|------------------|--------------------|
| electricity | 370 | 25,968 | 24 | 7 | $1.19e4$ | $7.99e3$ |
| traffic | 963 | 10,392 | 24 | 7 | $1.08e-2$ | $1.25e-2$ |
| wiki | 115,084 | 747 | 14 | 4 | $4.85e4$ | $1.26e4$ |
| PeMSD7(M) | 228 | 11,232 | 9 | 160 | 3.97 | 4.42 |

perform the forecast, without any need for re-training. We illustrate some representative results on a time-series from the dataset in Fig. 2. In Fig. 2a, we show some of the basis time-series (global features) extracted from the traffic dataset, which can be combined linearly to yield individual original time-series. It can be seen that the basis series are highly temporal and can be predicted in the test range using the network $\mathcal{T}_X(\cdot|\Theta_X)$. In Fig. 2b, we illustrate the complete architecture of DeepGLO. It can be observed that the output of the global TCN-MF model is inputted as a covariate to the TCN $\mathcal{T}_Y(\cdot)$, which inturn combines this with local features, and predicts in the test range through multi-step look-ahead predictions.

Table 2: Comparison of algorithms on normalized and unnormalized versions of data-sets on rolling prediction tasks The error metrics reported are WAPE/MAPE/SMAPE (see Section C.2). TRMF is retrained before every prediction window, during the rolling predictions. All other models are trained once on the initial training set and used for further prediction for all the rolling windows. Note that for DeepAR, the normalized column represents model trained with `scaler=True` and unnormalized represents `scaler=False`. Prophet could not be scaled to the wiki dataset, even though it was parallelized on a 32 core machine. Below the main table, we provide MAE/MAPE/RMSE comparison with the models implemented in [29], on the PeMSD7(M) dataset.

| | Algorithm | electricity $n = 370$ | | traffic $n = 963$ | | wiki $n = 115,084$ | |
|---|---|---|---|---|---|---|---|
| | | Normalized | Unnormalized | Normalized | Unnormalized | Normalized | Unnormalized |
| Proposed | DeepGLO | 0.133/0.453/0.162 | **0.082**/0.341/**0.121** | 0.166/ 0.210 /0.179 | **0.148/0.168/0.142** | 0.569/3.335/1.036 | 0.237/0.441/0.395 |
| | Local TCN (LeveledInit) | 0.143/0.356/0.207 | 0.092/**0.237**/0.126 | 0.157/**0.201**/0.156 | 0.169/0.177/0.169 | **0.243/0.545/0.431** | **0.212/0.316/0.296** |
| | Global TCN-MF | 0.144/0.485/0.174 | 0.106/0.525/0.188 | 0.336/0.415/0.451 | 0.226/0.284/0.247 | 1.19/8.46/1.56 | 0.433/1.59/0.686 |
| Local-Only | LSTM | 0.109/0.264/0.154 | 0.896/0.672/0.768 | 0.276/0.389/0.361 | 0.270/0.357/0.263 | 0.427/2.170/0.590 | 0.789/0.686/0.493 |
| | DeepAR | **0.086/0.259/ 0.141** | 0.994/0.818/1.85 | **0.140/0.201/ 0.114** | 0.211/0.331/0.267 | 0.429/2.980/0.424 | 0.993/8.120/1.475 |
| | TCN (no LeveledInit). | 0.147/0.476/0.156 | 0.423/0.769/0.523 | 0.204/0.284/0.236 | 0.239/0.425/0.281 | 0.336/1.322/0.497 | 0.511/0.884/0.509 |
| | Prophet | 0.197/0.393/0.221 | 0.221/0.586/0.524 | 0.313/0.600/0.420 | 0.303/0.559/0.403 | - | - |
| Global-Only | TRMF (retrained) | 0.104/0.280/0.151 | 0.105/0.431/0.183 | 0.159/0.226/ 0.181 | 0.210/ 0.322/ 0.275 | 0.309/0.847/0.451 | 0.320/0.938/0.503 |
| | SVD+TCN | 0.219/0.437/0.238 | 0.368/0.779/0.346 | 0.468/0.841/0.580 | 0.329/0.687/0.340 | 0.697/3.51/0.886 | 0.639/2.000/0.893 |

| Algorithm | PeMSD7(M) (MAE/MAPE/RMSE) |
|-----------|--------------------------|
| DeepGLO (Unnormalized) | **3.53/ 0.079 / 6.49** |
| DeepGLO (Normalized) | 4.53/ 0.103 / 6.91 |
| STGCN(Cheb) | 3.57/0.087/6.77 |
| STGCN($1^{st}$) | 3.79/0.091/7.03 |

# 6 Empirical Results

In this section, we validate our model on four real-world data-sets on rolling prediction tasks (see Section C.1 for more details) against other benchmarks. The data-sets in consideration are, $(i)$ electricity [23] - Hourly load on 370 houses. The training set consists of $25,968$ time-points and the task is to perform rolling validation for the next 7 days (24 time-points at a time, for 7 windows) as done in [29, 20, 9];$(ii)$ traffic [7] - Hourly traffic on 963 roads in San Francisco. The training set consists of $10m392$ time-points and the task is to perform rolling validation for the next 7 days (24 time-points at a time, for 7 windows) as done in [29, 20, 9] and $(iii)$ wiki [14] - Daily web-traffic on about $115,084$ articles from Wikipedia. We only keep the time-series without missing values from the original data-set. The values for each day are normalized by the total traffic on that day across all time-series and then multiplied by $1e8$. The training set consists of 747 time-points and the task is to perform rolling validation for the next 86 days, 14 days at a time. More data statistics indicating scale variations are provided in Table 1. $(iv)$ PeMSD7(M) [6] - Data collected from Caltrain PeMS system, which contains data for 228 time-series, collected at 5 min interval. The training set consists of 11232 time-points and we perform rolling validation for the next 1440 points, 9 points at a time.

For each data-set, all models are compared on two different settings. In the first setting the models are trained on *normalized* version of the data-set where each time series in the training set is re-scaled as $\tilde{\mathbf{y}}_{1:t-\tau}^{(i)} = (\mathbf{y}_{1:t-\tau}^{(i)} - \mu(\mathbf{y}_{1:t-\tau}^{(i)}))/(\sigma(\mathbf{y}_{1:t-\tau}^{(i)}))$ and then the predictions are scaled back to the original scaling. In the second setting, the data-set is *unnormalized* i.e left as it is while training and prediction. Note that all models are compared in the test range on the original scale of the data. The purpose of these two settings is to measure the impact of scaling on the accuracy of the different models.

The models under contention are: **(1)** DeepGLO: The combined local and global model proposed in Section 5.2. We use time-features like time-of-day, day-of-week etc. as global covariates, similar to DeepAR. For a more detailed discussion, please refer to Section C.3. **(2)** Local TCN (LeveledInit): The temporal convolution based architecture discussed in Section 4, with LeveledInit. **(3)** LSTM: A simple LSTM block that predicts the time-series values as function of the hidden states [11]. **(4)** DeepAR: The model proposed in [9]. **(5)** TCN: A simple Temporal Convolution model as described in Section 4. **(6)** Prophet: The versatile forecasting model from Facebook based on classical techniques [8]. **(7)** TRMF: the model proposed in [29]. Note that this model needs to be retrained for every rolling prediction window. **(8)** SVD+TCN: Combination of SVD and TCN. The original data is factorized as $\mathbf{Y} = \mathbf{U}\mathbf{V}$ using SVD and a leveled network is trained on the $\mathbf{V}$. This is a simple baseline for a global-only approach. **(9)** STGCN: The spatio-temporal models in [28]. We use the same hyper-parameters for DeepAR on the traffic and electricity datasets, as specified in [9], as implemented in GluonTS [1]. The WAPE values from the original paper could not be directly used, as there are different values reported in [9] and [20]. Note that for DeepAR, normalized and unnormalized settings corresponds to using `sclaing=True` and `scaling=False` in the GluonTS package. The model in TRMF [29] was trained with different hyper-parameters (larger rank) than in the original paper and therefore the results are slightly better. More details about all the hyper-parameters used are provided in Section C. The rank $k$ used in electricity, traffic, wiki and PeMSD7(M) are $64/60, 64/60, 256/1,024$ and $64/-$ for DeepGLO/TRMF.

In Table 2, we report WAPE/MAPE/SMAPE (see definitions in Section C.2) on the first three datasets under both normalized and unnormalized training. We can see that DeepGLO features among the top two models in almost all categories, under all three metrics. DeepGLO does better than the individual local TCN (LeveledInit) method and the global TCN-MF model on average, as it is aided by both global and local factors. The local TCN (LeveledInit) model performs the best on the larger wiki dataset with $> 30\%$ improvement over all models (not proposed in this paper) in terms of WAPE, while DeepGLO is close behind with greater than $25\%$ improvement over all other models. We also find that DeepGLO performs better in the unnormalized setting on all instances, because there is no need for scaling the input and rescaling back the outputs of the model. We find that the TCN (no LeveledInit), DeepAR[1] and the LSTM models do not perform well in the unnormalized setting as expected. In the second table, we compare DeepGLO with the models in [28], which can capture global features but require a weighted graph representing closeness relations between the time-series as *external* input. We see that DeepGLO (unnormalized) performs the best on all metrics, even though it does not require any external input. Our implementation can be found at https://github.com/rajatsen91/deepglo.

## Footnotes

[1]Note that for DeepAR, normalized means the GluonTS implementation run with `scaler=True` and unnormalized means `scaler=False`

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
