[Supplementary Material · DTRMF_sup.pdf]



Figure 3: Illustration of DLN

# A   Deep Leveled Network

Large scale time-series datasets containing upwards of hundreds of thousands of time-series can have very diverse scales. The diversity in scale leads to issues in training deep models, both Temporal Convolutions and LSTM based architectures, and some normalization is needed for training to succeed [16, 3, 9]. However, selecting the correct normalizing factor for each time-series is not an exact science and can have effects on predictive performance. For instance in [9] the data-sets are whitened using the training standard deviation and mean of each time-series while training, and the predictions are renormalized. On the other hand in [3], each time-series is rescaled by the value of that time-series on the first time-step. Moreover, when performing rolling predictions using a pre-trained model, when new data is observed there is a potential need for updating the scaling factors by incorporating the new time-points. In this section we propose DLN , a simple *leveling network* architecture that can be trained on diverse datasets without the need for a priori normalization.

DLN  consists of two temporal convolution blocks (having the same dynamic range/look-back $l$) that are trained concurrently. Let us denote the two networks and the associated weights by $\mathcal{T}_m(\cdot|\Theta_m)$ and $\mathcal{T}_r(\cdot|\Theta_r)$ respectively. The key idea is to have $\mathcal{T}_m(\cdot|\Theta_m)$ (the leveling component) to predict the rolling mean of the next $w$ future time-points given the past. On the other-hand $\mathcal{T}_r(\cdot|\Theta_r)$ (the residual component) will be used to predict the variations with respect to this mean value. Given an appropriate window size $w$ the rolling mean stays stable for each time-series and can be predicted by a simple temporal convolution model and given these predictions the additive variations are relatively scale free i.e. the network $\mathcal{T}_r(\cdot|\Theta_r)$ can be trained reliably without normalization. This can be summarized by the following equations:

$$[\hat{y}_{j-l+1}, \cdots, \hat{y}_j] = \mathcal{T}_{\mathsf{DLN}}\left(\mathbf{y}_{j-l:j-1}|\Theta_m, \Theta_r\right) := \mathcal{T}_m\left(\mathbf{y}_{j-l:j-1}|\Theta_m\right) + \mathcal{T}_r(\mathbf{y}_{j-l:j-1}|\Theta_r) \quad (4)$$

$$[\hat{m}_{j-l+1}, \cdots, \hat{m}_j] = \mathcal{T}_m(\mathbf{y}_{j-l:j-1}|\Theta_m) \quad (5)$$

$$[\hat{r}_{j-l+1}, \cdots, \hat{r}_j] = \mathcal{T}_r(\mathbf{y}_{j-l:j-1}|\Theta_r) \quad (6)$$

where we want $\hat{m}_j$ to be close to $\mu(\mathbf{y}_{j:j+w-1})$ and $\hat{r}_j$ to be close to $y_j - \mu(\mathbf{y}_{j:j+w-1})$. An illustration of the leveled network methodology is shown in the above figure.

**Training:** Both the networks can be trained concurrently given the training set $\mathbf{Y}^{(\text{tr})}$, using mini-batch stochastic gradient updates. The pseudo-code for training a DLN  is described in Algorithm 1. The loss function $\mathcal{L}(\cdot, \cdot)$ used is the same as the metric defined in Eq. (1). Note that in Step 9, the leveling component $\mathcal{T}_m(\cdot|\Theta_m)$ is held fixed and only $\mathcal{T}_r(\cdot|\Theta_r)$ is updated.

**Prediction:** The trained model can be used for multi-step look-ahead prediction in a standard manner. Given the past data-points of a time-series, $\mathbf{y}_{j-l:j-1}$, the prediction for the next time-step is given by $\hat{y}_j$ defined in (4). Now, the one-step look-ahead prediction can be concatenated with the past values to form the sequence $\tilde{\mathbf{y}}_{j-l+1:j} = [\mathbf{y}_{j-l+1:j-1}\hat{y}_j]$, which can be again passed through the network and get the next prediction: $[\cdots, \hat{y}_{j+1}] = \mathcal{T}_{\mathsf{DLN}}(\tilde{\mathbf{y}}_{j-l+1:j})$. The same procedure can be repeated $\tau$ times to predict $\tau$ time-steps ahead in the future. Table 3 shows the performance of leveled network on the same datasets and mertics used in Table 2.

Table 3: Performance of DLN  on the same datasets and metrics as in Table 2.

| | Algorithm | electricity $n = 370$ | | traffic $n = 963$ | | wiki $n = 115,084$ | |
| --- | --- | --- | --- | --- | --- | --- | --- |
| | | Normalized | Unnormalized | Normalized | Unnormalized | Normalized | Unnormalized |
| Proposed | Local DLN | 0.086/0.258/0.129 | 0.118/ 0.336/0.172 | 0.169/0.246/0.218 | 0.237/0.422/0.275 | 0.235/0.469/ 0.346 | 0.288/0.397/0.341 |

We will now provide a proof of Proposition 1.

*Proof of Proposition 1.* We will follow the convention of numbering the input layer as layer 0 and each subsequent layers in increasing order. 1a shows a TCN with the last layered numbered $d = 4$. Each neuron in a layer is numbered starting at 0 from the right-hand side. $a_{i,j}$ denote neuron $j$ in layer $i$. We will focus on the neurons in each layer that take part in the prediction $\hat{y}_j$. Note that on layer $d - 1$ ($d$ being the last layer), the neurons that are connected to the output $\hat{y}_j$, are $a_{d-1,0}$ and $a_{d-1,2^{d-1}}$. Therefore, we have $\hat{y}_j = \frac{1}{2}(a_{d-1,0} + a_{d-1,2^{d-1}})$, whenever the neurons have values greater equal to zero. Similary, any neuron $a_{l,s*2^l}$ is the average of $a_{l-1,s*2^l}$ and $a_{l-1,(2s+1)*2^{l-1}}$. Therefore, by induction we have that,

$$\hat{y}_j = \frac{1}{2 * 2^{d-1}} \sum_{i=1}^{2*2^{d-1}} y_{j-i}. \tag{7}$$

$\square$

## B  Rolling Prediction without Retraining

Once the TCN-MF model is trained using Algorithm 2, we can make predictions on the test range using multi-step look-ahead prediction. The method is straight-forward - we first use $\mathcal{T}_X(\cdot)$ to make multi-step look-ahead prediction on the basis time-series in $\mathbf{X}^{(\text{tr})}$ as detailed in Section 4, to obtained $\hat{\mathbf{X}}^{(\text{te})}$; then the original time-series predictions can be obtained by $\hat{\mathbf{Y}}^{(\text{te})} = \mathbf{F}\hat{\mathbf{X}}^{(\text{te})}$. This model can also be adapted to make rolling predictions *without retraining*. In the case of rolling predictions, the task is to train the model on a training period say $\mathbf{Y}[:, 1 : t_1]$, then make predictions on a future time period say $\hat{\mathbf{Y}}[:, t_1 + 1 : t_2]$, then receive the actual values on the future time-range $\mathbf{Y}[:, t_1 + 1 : t_2]$ and after incorporating these values make further predictions for a time range further in the future $\hat{\mathbf{Y}}[:, t_2 + 1 : t_3]$ and so on. The key challenge in this scenario, is to incorporate the newly observed values $\mathbf{Y}[:, t_1 + 1 : t_2]$ to generate the values of the basis time-series in that period which is $\mathbf{X}[:, t_1 + 1 : t_2]$. We propose to obtain these values by minimizing global loss defined in (3) while keeping $\mathbf{F}$ and $\mathcal{T}_X(\cdot)$ fixed:

$$\mathbf{X}[:, t_1 + 1 : t_2] = \underset{\mathbf{M} \in \mathbb{R}^{k \times (t_2 - t_1)}}{\arg\min} \mathcal{L}_G\left(\mathbf{Y}[:, t_1 + 1 : t_2], \mathbf{F}, \mathbf{M}, \mathcal{T}_X\right).$$

Once we obtain $\mathbf{X}[:, t_1 + 1 : t_2]$, we can make predictions in the next set of future time-periods $\hat{\mathbf{Y}}[:, t_2 + 1 : t_3]$. Note that the TRMF model in [29] needed to be retrained from scratch to incorporate the newly observed values. In this work retraining is not required to achieve good performance, as we shall see in our experiments in Section 6.

## C  More Experimental Details

We will provide more details about the experiments like the exact rolling prediction setting in each data-sets, the evaluation metrics and the model hyper-parameters.

### C.1  Rolling Prediction

In our experiments in Section 6, we compare model performances on rolling prediction tasks. The goal in this setting is to predict future time-steps in batches as more data is revealed. Suppose the initial training time-period is $\{1, 2, ..., t_0\}$, rolling window size $\tau$ and number of test windows $n_w$. Let $t_i = t_0 + i\tau$. The rolling prediction task is a sequential process, where given data till last window, $\mathbf{Y}[:, 1 : t_{i-1}]$, we predict the values for the next future window $\hat{\mathbf{Y}}[:, t_{i-1} + 1 : t_i]$, and then the actual values for the next window $\mathbf{Y}[:, t_{i-1} + 1 : t_i]$ are revealed and the process is carried on for $i = 1, 2, ..., n_w$. The final measure of performance is the loss $\mathcal{L}(\hat{\mathbf{Y}}[:, t_0 + 1 : t_{n_w}], \mathbf{Y}[:, t_0 + 1 : t_{n_w}])$ for the metric $\mathcal{L}$ defined in Eq. (1).

For instance, In the traffic data-set experiments we have $t_0 = 10392, \tau = 24, w = 7$ and in electricity $t_0 = 25968, \tau = 24, w = 7$. The wiki data-set experiments have the parameters $t_0 = 747, \tau = 14, w = 4$.

### C.2  Loss Metrics

The following well-known loss metrics [13] are used in this paper. Here, $\mathbf{Y} \in \mathbb{R}^{n' \times t'}$ represents the actual values while $\hat{\mathbf{Y}} \in \mathbb{R}^{n' \times t'}$ are the corresponding predictions.

**(i) WAPE:** Weighted Absolute Percent Error is defined as follows,

$$\mathcal{L}(\hat{\mathbf{Y}}, \mathbf{Y}) = \frac{\sum_{i=1}^{n'} \sum_{j=1}^{t'} |Y_{ij} - \hat{Y}_{ij}|}{\sum_{i=1}^{n'} \sum_{j=1}^{t'} |Y_{ij}|}. \tag{8}$$

**(ii) MAPE:** Mean Absolute Percent Error is defined as follows,

$$\mathcal{L}_m(\hat{\mathbf{Y}}, \mathbf{Y}) = \frac{1}{Z_0} \sum_{i=1}^{n'} \sum_{j=1}^{t'} \frac{|Y_{ij} - \hat{Y}_{ij}|}{|Y_{ij}|} \mathbb{1}\{|Y_{ij}| > 0\}, \tag{9}$$

where $Z_0 = \sum_{i=1}^{n'} \sum_{j=1}^{t'} \mathbb{1}\{|Y_{ij}| > 0\}$.

**(iii) SMAPE:** Symmetric Mean Absolute Percent Error is defined as follows,

$$\mathcal{L}_s(\hat{\mathbf{Y}}, \mathbf{Y}) = \frac{1}{Z_0} \sum_{i=1}^{n'} \sum_{j=1}^{t'} \frac{2|Y_{ij} - \hat{Y}_{ij}|}{|Y_{ij} + \hat{Y}_{ij}|} \mathbb{1}\{|Y_{ij}| > 0\}, \tag{10}$$

where $Z_0 = \sum_{i=1}^{n'} \sum_{j=1}^{t'} \mathbb{1}\{|Y_{ij}| > 0\}$.

### C.3 Model Parameters and Settings

In this section we will describe the compared models in more details. For a TC network the important parameters are the kernel size/filter size, number of layers and number of filters/channels per layer. A network described by $[c_1, c_2, c_3]$ implies that there are three layers with $c_i$ filters in layer $i$. For, an LSTM the parameters $(n_h, n_l)$ means that the number of neurons in hidden layers is $n_h$, and number of hidden layers is $n_l$. All models are trained with early stopping with a tenacity or patience of 7, with a maximum number of epochs 300. The hyper-parameters for all the models are as follows,

**DeepGLO:** In all the datasets, except wiki and PeMSD7(M) the networks $\mathcal{T}_X$ and $\mathcal{T}_Y$ both have parameters $[32, 32, 32, 32, 32, 1]$ and kernel size is 7. On the wiki dataset, we set the networks $\mathcal{T}_X$ and $\mathcal{T}_Y$ with parameters $[32, 32, 32, 32, 1]$. On the PeMSD7(M) dataset we set the parameters as $[32, 32, 32, 32, 32, 1]$ and $[16, 16, 16, 16, 16, 1]$. We set $\alpha$ and $\lambda_\mathcal{T}$ both to 0.2 in all experiments. The rank $k$ used in electricity, traffic, wiki and PeMSD7(M) are 64, 64 ,256 and 64 respectively. We use 7 time-covariates, which includes minute of the hour, hour of the day, day of the week, day of the month, day of the year, month of the year, week of the year, all normalized in a range $[-0.5, 0.5]$, which is a subset of the time-covariates used by default in the GluonTS library.

**Local TCN (LeveledInit):** We use the setting $[32, 32, 32, 32, 32, 1]$ for all datasets.

**Local DLN:** In all the datasets, the leveled networks have parameters $[32, 32, 32, 32, 32, 1]$ and kernel size is 7.

**TRMF:** The rank $k$ used in electricity, traffic, wiki are $60, 60$ ,1024 respectively. The lag indices are set to include the last day and the same day in the last week for traffic and electricity data.

**SVD+Leveled:** The rank $k$ used in electricity, traffic and wiki are $60, 60$ and 500 respectively. In all the datasets, the leveled networks have parameters $[32, 32, 32, 32, 32, 1]$ and kernel size is 7.

**LSTM:** In all datasets the parameters are $(45, 3)$.

**DeepAR:** In all datasets, we use the default parameters in the `DeepAREstimator` in the GluonTS implementation.

**TCN:** In all the datasets, the parameters are $[32, 32, 32, 32, 32, 1]$ and kernel size is 7.

**Prophet:** The parameters are selected automatically. The model is parallelized over 32 cores. The model was run with growth = 'logistic', as this was found to perform the best.

Note that we replicate the exact values reported in [28] for the STGCN models on the PeMSD7(M) dataset.