[Reviews · NeurIPS 2019]

Reviewer 1



Originality: This paper has a number of original ideas. It clearly points out that many recent forecasting papers with neural networks train on many series but focus forecasting on individual series. Instead, their novel contribution is to blend these two approaches using an attention mechanism. This attention mechanism simply determines if the forecaster should weight the global forecasting from temporal matrix factorization or the local forecasts trained on individual series. The entire thing is trained jointly so the local forecasts are mostly trained on sections where the local forecasts are important etc. The authors also introduce a method for dealing with the scale problem in forecasting by separately forecasting the mean and residuals. One question I had, which would be nice to see addressed, is why the attention mechanism is only based on the Y series, rather than both the Y series and the latent factors X + F. I could imagine that there could be other info included in X + F about how good global forecasts would be for a specific series. Clarity: good, I found it very easy to follow and understand. Significance: High. This is important. Many neural network architectures are build for text/audio/images etc but these cannot simply be ported over to high dimensional time series forecasting. The area needs its own original contributions, and this paper does just that. Quality: High. Experiments are well done, the methodology is well explained and well motivated.

Reviewer 2



1. The originality of this paper is not much. The two models proposed in this paper are the deep level network (DLN) and the hybrid model DeepGLO. The DLN is just a combination of two temporal convolution networks. One network models the rolling mean of a time series and the other one models the residual part. The DeepGLO simply combines a global model and a local model by a linear combination. Both of the ideas are not very original. 2. The authors emphasize the difficulty of normalizing data and show that they are dealing with this problem. However, the proposed model does not completely solve the problem. First, the proposed DLN only separates the mean and residual of time series, but the scale of the mean and the residual can still be large and is still a problem for neural networks. Therefore, the proposed model does not have the ability to solve the problem. Second, as shown in Table 2, the experimental results on the unnormalized data are not significantly better than those on the normalized data. Consequently, the authors are suggested to reconsider whether they should emphasize that they have solved the problem of normalization. 3. The organization of Section 4 and 5 is hard to follow. The reason could be that it is written in a bottom-up manner. The paper first introduces the local model, the global model and then the hybrid of the two. The problem is that some symbols in Algorithms 2 is not yet introduced when readers are reading that part, which may confuse the readers.The authors are suggested to move Section 5.2 forward. That is, introduce the whole model before explaining each part of the model. This will make the methodology clearer. 4. The sizes of the figures, tables and the font in the captions are too small, and they are difficult to read (Although this does not affect the current evaluation of this paper). It would be better to make the figures larger and just use the normal font size for captions and tables.

Reviewer 3



This paper proposed a hybrid model, which tries to tackle the challenges of multi-dimensional time series forecasting (TSF) from two perspectives: (1) to grasp the 'global' evolutionary laws of time series datasets that have a wide variation in scales of the individual time series; (2) to extract the 'local' patterns through a data-dependent attention model. The motivations are clear. The experimental results on 4 real-life tasks validate their superiority in comparison with other competing TSF methods. However, there is still a lack of necessary validation and explanations, especially with regard to the specific roles of the global and local models in TSF. Is it possible to provide some more detailed theoretical proofs or design a more detailed experiment on a dataset to demonstrate the power of global and local models? Otherwise, the contributions are weak and unclear. Besides, It would be better to extensively compare the empirical performance of DeepGLO with other existing hybrid models, such as Yu et al. Spatio-temporal Graph Convolutional Networks: A deep learning framework to traffic forecasting, 2018.

[Author Response · NeurIPS 2019]

**Rev#1:** We thank the reviewer for the encouraging remarks.

*Why the attention mechanism is only based on the Y series..... latent factors X,F?:* This is a good observation indeed
and we have thought of possible extensions in this direction. One method could be to treat the prediction from the local
model $\hat{\mathbf{y}}_t^l$ and the predicted basis time-series $\mathbf{X}_t$ as covariates for predicting the actual values $\mathbf{y}_t$ and feed them into a
temporal convolution network which is trained after the local and global models are trained.

*Details about what the loss function is specifically to train the mean and residual forecasters in the local models?:*
These details are implicitly specified in Algorithm 1. The loss function used to train the mean forecaster is the
normalized $\ell_1$-loss (see eq. 1) between the predicted value and the mean of next w values of the original time-series.
The residual forecaster is trained using the same loss, but with respect to the true residual values in the future time-range.
We will add a text description in our revised version.

**Rev#2:** We thank the reviewer for the comments and suggestions.

*Both the local and global models are not very original... :* To the best of our knowledge, this is the first paper
to propose a hybrid local and global model in the context of deep learning for time-series and we have a thorough
discussion of prior work, in the paper. The global model is especially novel, as matrix factorization regularized by
a temporal convolution network has not been attempted before. There are several differences from TRMF, such as
training the factors and the network alternatingly through SGD, unlike TRMF. The local model also highlights the
issue of normalization and diverse scales of different time-series, which is not commonly discussed in deep learning
time-series papers.

*The authors emphasize the difficulty of normalizing data and show that they are dealing with this problem ....:* We
would like to note that we emphasize the normalization issue as it is a commonly ignored. We propose the leveled
network method, where the key idea is that after the predicted rolling means (from the leveling network) is subtracted,
the remaining residual values have much less variation in scale and therefore the residual network can be trained more
reliably. As an empirical evidence, we would like to point the reviewer to the unnormalized columns in Table 2, where
we see that when the data is not normalized, the local only models like Temporal Conv., DeepAR, LSTM do not
converge to a good solution at all, while the Local DLN model performs at par with the normalized versions. Moreover,
we would like to point out that on the larger wiki dataset, the unnormalized versions of DeepGLO and local DLN
models perform better than the normalized versions in several metrics. We agree with the reviewer that the residual
values may also have variations among the time-series, however the variations are much less and therefore our proposed
solution empirically works well in all the data-sets considered.

*Organization of Section 4 and 5 and sizes of figures:* We thank the reviewer for these suggestions and they will be
incorporated into the revision of the paper.

**Rev#3**: We thank the reviewer for the comments and suggestions.

*However, there is still a lack ....especially with regard to the specific roles of the global and local models....:* In Table 2
from the paper, we separately provide the metrics from the hybrid DeepGLO model and the local only DLN model,
which shows an improvement of DeepGLO over the local only model. In Table 1 below, we further provide the metrics
from the global only DLN-MF model, in response to this question. We can see that the hybrid DeepGLO model is better
than the local and global counterparts in all cases, thus proving that there is added value in having a hybrid model. We
will add these additional results to the paper.

| Algorithm | elec $n = 370$ | | traffic $n = 963$ | | Algorithm | PeMSD7(M) | | |
| | Normalized | Unnormalized | Normalized | Unnormalized | | MAE | MAPE (%) | RMSE |
|---|---|---|---|---|---|---|---|---|
| DeepGLO | 0.084/0.291/0.119 | 0.109/0.448/0.149 | 0.159/0.218/0.202 | 0.169/0.256/ 0.195 | DeepGLO | 3.81 | **8.29** | **6.31** |
| Local DLN | 0.086/0.258/0.129 | 0.118/0.336/0.172 | 0.169/0.246/0.218 | 0.237/0.422/0.275 | STGCN(Cheb) | **3.57** | 8.69 | 6.77 |
| DLN-MF | 0.255/0.687/0.449 | 0.349/0.696/0.539 | 0.247/0.281/0.291 | 0.176/0.234/0.203 | STGCN($1^{st}$) | 3.79 | 9.12 | 7.03 |

Table 1: In the first table, we provide additional values for the global only DLN-MF part for DeepGLO. In the second table, we
compare the DeepGLO model with the Spatio-Temporal Model from Yu. et al.

*Comparison with Yu et al. Spatio-temporal Graph Convolutional Networks...:* We have already cited the above paper.
Following the request of the reviewer, we compare our DeepGLO model on the PeMSD7(M) dataset on the same test
split on the same task (of predicting 45 min in to the future) as in the original paper. The results are in Table 1 and
we can see that DeepGLO performs better in two metrics, even when DeepGLO does not have access to the weighted
similarity graph, which is an additional input to the model in Yu et al. We will add these comparisons to our paper. In
view of these new results, we hope that the reviewer reconsiders their rating of the paper.

[Meta-Review · NeurIPS 2019]

This paper proposes extending temporal matrix factorization to incorporate neural network regularization for time series forecasting. The intuition is to capture global and local structure to make forecasts. The work is interesting because it bridges more classical forecasting ideas with new state of the art deep learning approaches. The ideas presented in this paper seem novel as the authors take existing building blocks for deep learning and combine them in a creative way to capture interesting structure of time series. This is in contrast to simply applying an existing deep learning approach such as seq2seq. There were some criticisms of the experimental evaluation which the authors seem to have addressed in their response and will include new comparisons that the reviewers asked for. The outstanding concern is in regards to understanding what each component of the proposed model is contributing to the increased performance as the current experiments do not elucidate this. However, I think that the ideas presented in this paper are very interesting and that the improved forecasting abilities of the model are good enough for publication now with a deeper understanding being followup work.